# Reduced- or Half-Dose Rivaroxaban Following Left Atrial Appendage Closure: A Feasible Antithrombotic Therapy in Patients at High Risk of Bleeding?

**DOI:** 10.3390/jcm12030847

**Published:** 2023-01-20

**Authors:** Xiao-Dong Zhou, Qin-Fen Chen, Fang Lin, Liangguo Wang, Yihe Chen, Dongjie Liang, Weijian Huang, Fangyi Xiao

**Affiliations:** 1Department of Cardiology, Heart Center, First Affiliated Hospital of Wenzhou Medical University, Wenzhou 325000, China; 2Department of Medical and Health Care Center, First Affiliated Hospital of Wenzhou Medical University, Wenzhou 325000, China; 3Department of Cardiology, Key Laboratory of Cardiovascular Disease of Wenzhou, First Affiliated Hospital of Wenzhou Medical University, Wenzhou 325000, China

**Keywords:** atrial fibrillation, left atrial appendage closure, rivaroxaban, stroke, bleeding, device-related thrombosis

## Abstract

The optimal antithrombotic strategy after percutaneous left atrial appendage closure (LAAC) has not yet been established. The advisability of administering low-dose direct oral anticoagulation after LAAC to patients at high risk of bleeding is uncertain. Thus, in the present study, we evaluated the safety and effectiveness of reduced-(15 mg) or half-dose rivaroxaban (10 mg) versus warfarin regarding real-world risks of thromboembolism, bleeding, and device-related thrombosis (DRT) after LAAC. Patients with non-valvular atrial fibrillation and HASBLED ≥ 3 who had undergone successful LAAC device implantation from October 2014 to April 2020 were screened and those who had received 10 mg or 15 mg rivaroxaban or warfarin therapy were enrolled. The patients were followed up 45 days and 6 months after LAAC to evaluate outcomes, including death, thromboembolism, major bleeding, and DRT. Of 457 patients with HASBLED ≥ 3 who had undergone LAAC, 115 had received warfarin and 342 rivaroxaban (15 mg: N = 164; 10 mg: N = 178). There were no significant differences in the incidence of thromboembolism or DRT between the warfarin and both doses of rivaroxaban groups (all *p* > 0.05). The incidence of major bleeding was significantly higher in the warfarin group than in either the reduced- or half-dose rivaroxaban groups (warfarin vs. rivaroxaban 15 mg: 2.6% vs. 0%, *p* = 0.030; warfarin vs. rivaroxaban 10 mg: 2.6% vs. 0%, *p* = 0.038). Either reduced- or half-dose rivaroxaban may be an effective and safe alternative to warfarin therapy in patients with LAAC and who are at high risk of bleeding, the risk of thromboembolism being similar and of major bleeding lower for both doses of rivaroxaban.

## 1. Introduction

Left atrial appendage closure (LAAC) has been widely performed for thromboprophylaxis in patients with non-valvular atrial fibrillation (NVAF) [1]. Because LAAC and oral anticoagulants (OACs) are similarly effective at preventing ischemic stroke, LAAC is currently recommended as an alternative for patients with NVAF and who are at high risk of bleeding or with other contraindications to long-term anticoagulation treatment [2]. After LAAC, a short course of antithrombotic therapy should be prescribed to permit complete endothelialization over the device and prevent device-related thrombosis (DRT) [3]. However, the optimal post-LAAC antithrombotic strategy has not yet been clarified, leaving the choice of a variety of treatment options after device implantation. According to the findings of the PROTECT-AF, PREVAIL, CAP, and CAP_2_ studies [4,5,6], the currently recommended post-procedural antithrombotic therapies are warfarin and single-antiplatelet therapy (SAPT) for 45 days, dual-antiplatelet therapy (DAPT) for 6 months after documentation of satisfactory LAAC (peri-device leak < 5 mm), or lifetime aspirin therapy for patients without contraindications to OACs. Direct oral anticoagulation (DOAC) is also considered a feasible alternative to warfarin for the initial period.

In Asian countries, safety concerns based on the generally low body mass index of the population and their different susceptibility to bleeding have resulted in lower DOAC doses generally being preferred. The J-ROCKET AF trial demonstrated that low-dose rivaroxaban (10 or 15 mg) is equally effective and is associated with lower risks of major bleeding than warfarin is in Asian patients with NVAF [7]. Although DOAC is superior to warfarin regarding major bleeding and intracranial hemorrhages, minor bleeding is not uncommon in patients receiving full-dose DOAC; this prejudices patient adherence to full-dose DOAC. Therefore, considerable attention has been paid to the safety and effectiveness of low-dose rivaroxaban after LAAC in Asian individuals. Currently, evidence-based justification for the strategy of reduced-dose anticoagulant administration after LAAC is limited. In the EWOLUTION trial, 41% of patients received reduced-dose DOAC; however, this study failed to clarify the advisability of a reduced-dose DOAC strategy after LAAC [8]. Fu et al. [9] compared the safety and efficacy of warfarin and reduced DOAC (rivaroxaban 15 mg daily). However, the efficacy of half-dose should also be further investigated because most such patients also receive antiplatelet therapy.

Substituting low-dose (reduced- or half-dose) DOAC for warfarin can be reasonably assumed to provide an excellent balance between maintaining a low risk of thromboembolic events and minimizing the risk of bleeding events in Asian patients with AF after LAAC. Thus, in the present study, patients at high risk of bleeding who had undergone LAAC were allocated to one of the following groups: reduced-dose DOAC, half-dose DOAC, and standard warfarin therapy. We retrospectively analyzed the safety and effectiveness of reduced-dose (rivaroxaban 15 mg daily) and half-dose DOAC (rivaroxaban 10 mg daily) versus standard warfarin therapy (warfarin) in Asian patients with NVAF after successful percutaneous LAAC.

## 2. Methods

### 2.1. Study Cohort

Consecutive patients with NVAF who had undergone LAAC at the First Affiliated Hospital of Wenzhou Medical University (Wenzhou, China) from October 2014 to April 2020 were enrolled in this retrospective study.

Data were collected retrospectively from the online information systems of the hospital. The study conformed to the principles outlined in the Declaration of Helsinki.

### 2.2. LAAC Procedures

Both transesophageal echocardiography (TEE) and cardiac computed tomography angiography were performed as necessary to detect left atrial thrombi and evaluate the anatomic structure of the left atrial appendage (LAA) before the procedure. The details of the LAAC procedure have been previously reported [10]. All procedures were performed by two experienced electrophysiologists under echocardiographic and fluoroscopic guidance. The type of device was selected based on the anatomic structure of the LAA.

### 2.3. Post-Procedural Antithrombotic Strategies

The dose of anticoagulants was individualized at the physician’s discretion in accordance with the risk of bleeding. Specialist physicians prescribed 10 mg of rivaroxaban for patients who met any one of the following criteria: (1) age ≥ 75 years; (2) previous major bleeding issues of unknown or untreated etiology; (3) estimated glomerular filtration rate < 50 mL/min/1.73 m^2^; (4); or intolerance to 15 mg of rivaroxaban (R^15^) because of recurrent minor bleeding (such as skin or urinary tract bleeding). OAC therapy was usually continued for at least 45 days after successful LAAC, whereas DAPT (aspirin 100 mg plus clopidogrel 75 mg) was continued for 1.5–6 months. Lifelong SAPT was subsequently administered.

### 2.4. Definitions

The major primary endpoints were thromboembolic complications, hemorrhagic strokes, major bleeding events, and DRT during the first 6 months after the procedure. Thromboembolic complications included strokes, transient ischemic attacks, and systemic embolism. Ischemic and hemorrhagic strokes were defined as the presence of clinically relevant focal neurological symptoms with consistent abnormalities on computed tomography or magnetic resonance imaging confirmed by a neurologist. Major bleeding was classified as intracranial, retroperitoneal, intraspinal, intraocular, or pericardial hemorrhage as indicated by a decrease in hemoglobin concentrations of more than 2 g/dL or requiring transfusion of ≥2 units of packed red blood cells [11]. Standard warfarin therapy was defined as taking warfarin for at least 45 days after LAAC with a target INR between 2 and 3 and an individual therapeutic range >70%. TEE was performed on the forty-fifth to sixtieth day, and 6 and 12 months after the procedure. All echocardiograms were evaluated independently by two experienced readers using a commercially available imaging system and a consensus was reached regarding the diagnosis of DRT.

### 2.5. Statistical Analysis

Data were analyzed using SPSS version 23.0 (IBM, Armonk, NY, USA). Patients’ clinical characteristics are reported as descriptive statistics. To adjust for baseline differences between the rivaroxaban and warfarin groups, patients were matched for age, sex, CHA_2_DS_2_Vasc score, and HAS-BLED score. Data are presented as the mean ± standard derivation for continuous and normally distributed variables, and as frequencies (percentage) for categorical variables. Comparisons were performed using an independent-samples Student’s *t*-test and the χ^2^ test. Non-normally distributed variables are presented as the median (quartile range) and were compared using the Mann–Whitney U test. The criterion for statistical significance was considered to be *p* < 0.05.

## 3. Results

A study flow diagram is presented in Figure 1. In total, 750 patients with NVAF were screened for LAAC, 23 of whom were excluded because their LAA anatomy was unsuitable for the available occlusive devices. LAAC devices were successfully deployed in 709 (97.5%) of the 727 patients in whom implantation was attempted. A further 136 patients were excluded because they were receiving other anticoagulant therapies or had other devices implanted, 57 because they had HASBLED scores < 3, and 12 because they were lost to follow-up. The remaining patients, who had received warfarin with a therapeutic range > 70% or rivaroxaban 10 mg daily or rivaroxaban 15 mg daily, were considered eligible for the present study. Their baseline characteristics are presented in Table 1 and Table 2. There were no significant differences in baseline characteristics between the low-dose rivaroxaban and warfarin groups. 

### 3.1. Warfarin versus Low-Dose Rivaroxaban

All study patients were followed up by telephone or outpatient visits for the 2.7-year follow-up period. Details are shown in Table 3. All patients with the named events were receiving routine antithrombotic treatments. Four of the 12 patients who had ischemic strokes had documented thromboembolic events while taking OACs. The incidence of thromboembolism while taking OACs did not differ significantly between the low-dose rivaroxaban and warfarin groups (1.7% [95% CI 0.5–6.1%] vs. 1.2% [95% CI 0.5–3.0%]; *p* = 0.643). However, warfarin was associated with a higher incidence of major bleeding events than low-dose rivaroxaban was (2.6% [95% CI 0.9–7.4%] vs. 0% [95% CI 0–1.1%]; *p* = 0.003).

A total of 100 (87.0%) of 115 patients who were taking warfarin underwent TEE after 45 days, whereas 311 (90.9%) patients who were taking rivaroxaban completed TEE examination after implantation. DRT occurred in three patients while they were taking OACs; two of these were detected by routine TEE but not linked to clinical events, whereas the third patient had an ischemic stroke, with DRT being confirmed by TEE. The incidence of DRT while taking OACs was similar in the rivaroxaban and warfarin groups (0.9% [95% CI 0.2–4.8%] vs. 0.6% [95% CI 0.2–2.1%], *p* = 0.744). 

### 3.2. Warfarin versus Reduced-Dose or Half-Dose Rivaroxaban

As shown in Table 4, compared with the warfarin group, both reduced-dose and half-dose rivaroxaban groups had a similar incidence of thromboembolic events while taking OACs (warfarin vs. R^15^: 1.7% [95% CI 0.5–6.1%] vs. 1.2% [95% CI 0.3–4.0%]; *p* = 0.719 and warfarin vs. R^10^: 1.7% [95% CI 0.5–6.1%] vs. 1.2% [95% CI 0.3–4.3%]; *p* = 0.657) but a lower incidence of major bleeding (warfarin vs. R^15^: 2.6% [95% CI 0.9–7.4%] versus 0% [95% CI 0–2.3%]; *p* = 0.038 and warfarin vs. R^10^: 2.6% [95% CI 0.9–7.4%] vs. 0% [95% CI 0–2.1%]; *p* = 0.030). There were no statistically significant differences in the rate of DRT between the warfarin, reduced-dose and half-dose groups (warfarin vs. R^15^: 0.9% [95% CI 0.2–4.8%] vs. 0.6% [95% CI 0.1–3.3%]; *p* = 0.800 and warfarin vs. R^10^: 0.9% [95% CI 0.2–4.8%] vs. 0.6% [95% CI 0.1–3.1%]; *p* = 0.754).

## 4. Discussion

### 4.1. Main Findings

This retrospective study provides data regarding the efficacy and safety of reduced-dose and half-dose rivaroxaban therapy after LAAC device implantation in Chinese individuals. Compared with warfarin, both reduced-and half-dose rivaroxaban were associated with a lower incidence of major bleeding without compromising efficacy in preventing thromboembolism and DRT.

### 4.2. DOAC as an Alternative to Warfarin after LAAC

The main advantage of LAAC over OACs is a significant reduction in the risk of bleeding associated with long-term OACs in patients with NVAF [12]. LAAC is indicated for stroke prevention in patients with NVAF and a high risk of stroke who are intolerant of long-term anticoagulant treatment; selecting the optimal antithrombotic strategy after LAAC is challenging in such patients.

In Asian countries, safety concerns based on the generally low body mass index of the population and their different susceptibility to bleeding have resulted in lower DOAC doses generally being preferred [13,14]. A retrospective study compared the effectiveness and safety of rivaroxaban following ROCKET AF (20/15 mg/d) and J-ROCKET AF (15/10 mg/d) among Asians with NVAF [15,16]. These researchers found that the studied doses of rivaroxaban were as effective as warfarin in preventing thromboembolic events and were associated with a significantly lower risk of intracranial hemorrhage and major bleeding. In patients with an eGFR of <50 mL/min/1.73 m^2^, the risk of major bleeding was lower with the J-ROCKET AF dosage than with the ROCKET AF dosage. These findings support the use of reduced-dose rivaroxaban as an alternative to standard-dose rivaroxaban or warfarin for stroke prevention in Asian patients. In the present study, we demonstrated that our reduced-dose rivaroxaban strategy (which was close to the J-ROCKET AF dosage) may be similarly effective regarding stroke prevention and further minimization of bleeding risks in patients with LAAC.

### 4.3. Reduced-Dose or Half-Dose DOAC as an Alternative to Warfarin after LAAC

Several studies have assessed the feasibility and safety of DOAC after LAAC. Bergmann et al. [3] demonstrated that DOAC therapy for 3 months after implantation is safe and effective, with low incidences of DRT and bleeding events compared with post-procedural warfarin treatment (DRT: 1/123 with DOAC vs. 1/75 with warfarin; bleeding: 2/105 with DOAC vs. 7/146 with warfarin); however, only 59% of patients in that study received full-dose DOAC therapy. The effects of the type and dose of DOAC on outcomes have not been analyzed. A multicenter retrospective analysis demonstrated that, after LAAC with a Watchman device, DOAC therapy (comprising rivaroxaban in 46% of patients) is associated with a similar prevalence of thromboembolism as warfarin (1.4% vs. 0.9%; *p* > 0.99) without an increased bleeding risk (0.5% versus 0.9; *p* = 0.6) [8]; however, that study did not provide data regarding clinical outcomes stratified by DOAC dose. Subsequently, Gu et al. [17] suggested that, after LAAC using a Watchman device, standard-dose rivaroxaban may be an acceptable alternative therapy to warfarin, having a lower incidence of thrombotic complications and bleeding events in Chinese individuals. The ADRIFT study, a multicenter, phase IIb study in which 105 patients were randomized after successful LAAC to either rivaroxaban 10 mg, rivaroxaban 15 mg, or DAPT with aspirin 75 mg and clopidogrel 75 mg, demonstrated that the clinical endpoints did not differ significantly between groups, supporting an alternative to the antithrombotic regimens currently used after LAAC [18]. 

### 4.4. DOAC versus DAPT Following LAAC

DAPT is most commonly used in patients with contraindications to OACs [19]. In studies involving the implantation of Amplatzer or LAmbre devices, the main treatment after LAAC was short-term DAPT followed by long-term SAPT [20,21]. However, from a pathophysiological standpoint, OACs are more appropriate than DAPT for reducing the thrombotic risk following LAAC [22,23]. Whether antiplatelet therapy is directly associated with an increased risk of DRT remains controversial [18,23]. Even if DAPT is effective in preventing DRT, it remains associated with a substantial incidence of bleeding events [18]. The ACTIVE trial found that DAPT is associated with significantly higher incidences of stroke and non-central nervous system systemic embolism and a similar bleeding risk compared with warfarin [19]. Patients with contraindications to OACs who are treated with DAPT still have a significant bleeding risk, with an estimated annual bleeding rate of 6.6–14.4% in the initial phase [24,25,26,27,28]. A multicenter study of 592 patients with relative contraindications to long-term anticoagulation therapy who underwent LAAC and received either DAPT or DOAC for 1–3 months showed that DAPT is associated with higher incidences of early death (3.7% vs. 1.1%), non-procedural-related severe bleeding (7.4% vs. 3.2%), and serious adverse events (11.1% vs. 5.3%) [28]. Furthermore, a meta-analysis of 15 randomized controlled trials showed that DOAC is associated with lower incidences of stroke, major bleeding, DRT, and all-cause mortality than is antiplatelet therapy [29]. Most recently, Della Rocca et al. [30] performed a prospective, nonrandomized study of 555 patients with NVAF in which they evaluated the long-term efficacy of a standard antithrombotic strategy (DOAC plus aspirin for 45 days, DAPT until 6 months, and aspirin continuing indefinitely) versus half-dose DOAC after Watchman implantation. They found that a half-dose DOAC strategy (half-dose DOAC plus aspirin and long-term half-dose DOAC monotherapy thereafter) significantly reduces the risk of the composite endpoint of DRT, thrombotic events, and major bleeding events. However, termination of anticoagulation is one of the main aims in patients undergoing LAAC. The strategy of lifelong anticoagulation is not easily accepted by patients who are considering undergoing LAAC, especially Asian individuals for whom half-dose anticoagulation is widely prescribed.

The current evidence combined with our findings suggests that either reduced- or half-dose DOAC following LAAC may offer a promising efficacy and safety profile in Asian patients with NVAF. Further randomized controlled trials are needed to determine whether reduced-dose DOAC does indeed have clinical advantages.

## 5. Limitations

The generalizability of the present findings is limited by several factors. First, this was a retrospective, observational study with potential selection bias because the post-procedural drug type and dose were not randomized. Second, the limited number of cases and low incidences of DRT, ischemic stroke/transient ischemic attack/systemic embolism, and bleeding events after implantation may have prevented the differences between the groups from reaching statistical significance. Hence, larger prospectively designed studies are needed to provide high quality data on this topic. Third, this study only included patients treated with low-dose rivaroxaban or warfarin; thus, our findings are not necessarily applicable to other DOAC therapies.

## 6. Conclusions

The present findings are important for clinical practice because DOACs are being increasingly prescribed as a result of their favorable safety and efficacy profiles. Either reduced- or half-dose rivaroxaban may constitute an effective and safe alternative to warfarin in patients with NVAF who are at high risk of bleeding, providing a good balance between the incidences of thromboembolism and major bleeding events during endothelialization after LAAC. Future studies should specifically evaluate the optimal dose of rivaroxaban for balancing effectiveness and safety in Asian patients who have undergone LAAC.

## Figures and Tables

**Figure 1 jcm-12-00847-f001:**
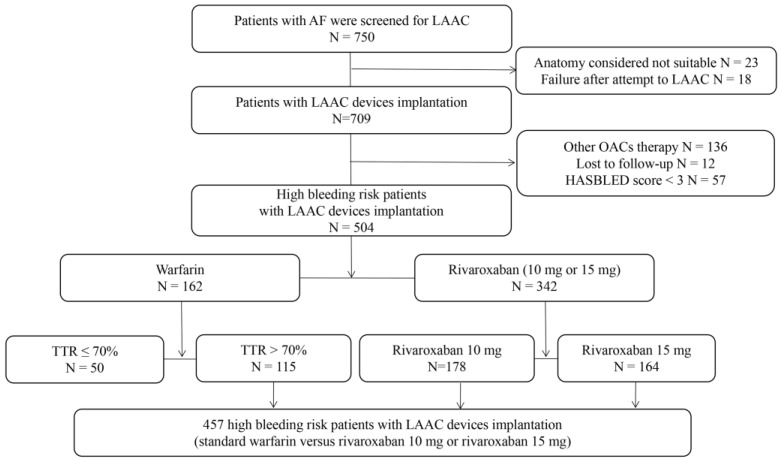
Patient flowchart. AF, atrial fibrillation; LAAC, left atrial appendage closure; OACs, oral anticoagulants; TTR, therapeutic range.

**Table 1 jcm-12-00847-t001:** Baseline patient characteristics stratified by warfarin and low-dose rivaroxaban.

	Warfarin N = 115	R N = 342	*p* Value Warfarin versus R
Age, years	71.7 ± 8.0	71.9 ± 7.2	0.802
Male	68 (61.7%)	229 (67.0%)	0.128
Non-paroxysmal AF	77 (67.0%)	206 (60.2%)	0.199
CHF	32 (27.8%)	70 (20.5%)	0.546
History of stroke	77 (67.0%)	218 (62.7%)	0.527
Hypertension	100 (87.0%)	277 (81.0%)	0.305
Diabetes	27 (23.5%)	104 (30.4%)	0.282
Vascular disease	42 (36.5%)	110 (32.1%)	0.309
eGFR, mL/min/1.73 m^2^	81.3 ± 12.8	80.3 ± 13.0	0.474
CHA_2_DS_2_-VASc score	4.6 ± 1.8	4.5 ± 1.5	0.558
HAS-BLED score	3.9 ± 1.2	3.8 ± 1.0	0.079
Type of LAAC device			
Watchman	63 (54.7%)	172 (50.3%)	0.405
ACP	29 (25.2%)	80 (23.4%)	0.691
LAmbre	23 (20.0%)	90 (26.3%)	0.174

Data are presented as mean ± SD, median (interquartile range) or N (%). Abbreviations: ACP, AMPLATZER cardiac plug; AF, atrial fibrillation; CHF, congestive heart failure; R, rivaroxaban.

**Table 2 jcm-12-00847-t002:** Baseline patient characteristics stratified by warfarin, reduced-dose, and half-dose rivaroxaban.

	Warfarin N = 115	R^10^ N = 178	R^15^ N = 164	*p* Value Warfarin versus R^10^	*p* Value Warfarin versus R^15^
Age, years	71.7 ± 8.0	72.8 ± 7.5	70.4 ± 7.6	0.233	0.073
Male	68 (61.7%)	124 (69.7%)	105 (64.0%)	0.640	0.407
Non-paroxysmal AF	77 (67.0%)	117 (65.7%)	89 (54.2%)	0.828	0.494
CHF	32 (27.8%)	40 (22.5%)	30 (18.2%)	0.299	0.059
History of stroke	77 (67.0%)	121 (59.9%)	103 (62.8%)	0.855	0.476
Hypertension	100 (87.0%)	139 (78.1%)	138 (84.1%)	0.056	0.514
Diabetes	27 (23.5%)	50 (28.1%)	54 (32.9%)	0.381	0.087
Vascular disease	42 (36.5%)	63 (35.3%)	47 (28.7%)	0.844	0.067
eGFR, mL/min/1.73 m^2^	81.3 ± 12.8	79.2 ± 13.0	82.2 ± 12.4	0.175	0.556
CHA_2_DS_2_-VASc score	4.6 ± 1.8	4.6 ± 1.5	4.3 ± 1.5	0.997	0.131
HAS-BLED score	3.9 ± 1.2	3.8 ± 1.0	3.7 ± 0.9	0.441	0.112
Type of LAAC device					
Watchman	63 (54.7%)	84 (47.2%)	88 (53.7%)	0.204	0.853
ACP	29 (25.2%)	48 (27.0%)	32 (19.5%)	0.740	0.256
LAmbre	23 (20.0%)	46 (25.8%)	44 (26.8%)	0.250	0.189

R^10^, rivaroxaban 10 mg; R^15^, rivaroxaban 15 mg; ACP, AMPLATZER cardiac plug; AF, atrial fibrillation; CHF, congestive heart failure.

**Table 3 jcm-12-00847-t003:** Clinical outcomes stratified by warfarin and low-dose rivaroxaban.

Variables	Warfarin N = 115	R N = 342	*p* Value Warfarin versus R
Thromboembolism events	4 (3.5%)	8 (2.3%)	0.509
Under OAC	2 (1.7%)	4 (1.2%)	0.643
Under DAPT	1 (0.9%)	2 (0.6%)	0.744
Under SAPT	1 (0.9%)	2 (0.6%)	0.744
Major bleeding	5 (4.3%)	3 (0.9%)	0.014
Under OAC	3 (2.6%)	0 (0%)	0.003
Under DAPT	1 (0.9%)	2 (0.6%)	0.744
Under SAPT	1 (0.9%)	1 (0.3%)	0.417
Device related thrombus	3 (2.6%)	3 (0.9%)	0.158
Under OAC	1 (0.9%)	2 (0.6%)	0.744
Under DAPT	2 (1.7%)	0 (0%)	0.015
Under SAPT	0 (0%)	1 (0.3%)	0.562
Resident leak at 45d TEE	100 (87.0%)	311 (91.6%)	0.220
<1 mm	85 (85.0%)	270 (87.1%)	0.262
1–3 mm	13 (13.0%)	34 (9.9%)	0.677
3–5 mm	2 (2.0%)	5 (1.5%)	0.834
>5 mm	0 (0%)	2 (1.2%)	0.411

Data are presented as mean ± SD, median (interquartile range), or N (%). Abbreviations: OAC, oral anticoagulants; DAPT, dual antiplatelet therapy; SAPT, single antiplatelet therapy; R, rivaroxaban; TEE, transesophageal echocardiography.

**Table 4 jcm-12-00847-t004:** Clinical outcomes in patients stratified by warfarin, reduced-dose rivaroxaban, and half-dose rivaroxaban.

Variables	Warfarin N = 115	R^10^ N = 178	R^15^ N = 164	*p* Value Warfarin versus R^10^	*p* Value Warfarin versus R^15^
Thromboembolism events	4 (3.5%)	4 (2.2%)	4 (1.8%)	0.528	0.609
Under OAC	2 (1.7%)	2 (1.2%)	2 (1.2%)	0.657	0.719
Under DAPT	1 (0.9%)	1 (0.6%)	1 (0.6%)	0.755	0.800
Under SAPT	1 (0.9%)	1 (0.6%)	1 (0.6%)	0.755	0.800
Major bleeding	5 (4.3%)	1 (0.6%)	2 (1.2%)	0.025	0.200
Under OAC	3 (2.6%)	0 (0%)	0 (0%)	0.030	0.038
Under DAPT	1 (0.9%)	1 (0.6%)	1 (0.6%)	0.755	0.800
Under SAPT	1 (0.9%)	0 (0%)	1 (0.6%)	0.213	0.800
Device-related thrombus	3 (2.6%)	1 (0.6%)	2 (1.2%)	0.140	0.389
Under OAC	1 (0.9%)	1 (0.6%)	1 (0.6%)	0.754	0.800
Under DAPT	2 (1.7%)	0 (0%)	0 (0%)	0.077	0.090
Under SAPT	0 (0%)	0 (0%)	1 (0.6%)	1.000	0.402
Resident leak	100 (87.0%)	163 (91.6%)	148 (90.2%)	0.203	0.389
<1 mm	85 (85.0%)	142 (87.1%)	128 (86.5%)	0.628	0.742
1–3 mm	13 (13.0%)	19 (11.7%)	15 (9.1%)	0.746	0.970
3–5 mm	2 (2.0%)	1 (0.6%)	4 (2.7%)	0.320	0.350
>5 mm	0 (0%)	1 (0.6%)	1 (0.7%)	0.433	0.780

R^10^, rivaroxaban 10 mg; R^15^, rivaroxaban 15 mg; OAC, oral anticoagulants; DAPT, dual antiplatelet therapy; SAPT, single antiplatelet therapy.

## Data Availability

The data that support the findings of this study are available from the first author (zhouxiaodong@wmu.edu.cn) upon reasonable request.

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
