# Peer review of "Reduced- or Half-Dose Rivaroxaban Following Left Atrial Appendage Closure: A Feasible Antithrombotic Therapy in Patients at High Risk of Bleeding?"

_jcm, 2023, doi:10.3390/jcm12030847_

Round 1

Reviewer 1 Report

The header could be a little more “catchy”, triggering

Low dose rivaroxaban was off label at the time of the study?

Which device were sued over time and in both groups comparatively?

Please provided not just TTR but INR range

CI are suggested. I am not so sure whether the conclusions would stand a larger trial. Therefore, more careful wording is suggested.

Table 3/4: how many patients were on OAC and APT? any information about OAC APT at time of event? Temporal distribution of events from time of device implantations might be helpful.

The paper would benefit from some rewording, in particular the part highlighted in yellow

Author Response

Reviewer 1
Point 1: The header could be a little more “catchy”, triggering

Response 1: Thank you for this suggestion, in response to which we have revised the manuscript title as follows: “Reduced- or half-dose rivaroxaban following left atrial appendage closure: a feasible antithrombotic therapy in patients at high risk of bleeding?”

Point 2: Low dose rivaroxaban was off label at the time of the study?

Response 2: Thank you for this comment. According to both previous and current international and Chinese guidelines, 15mg rivaroxaban is on-label, whereas 10mg rivaroxaban is off-label. The use of low dose rivaroxaban was individualized at the physician’s discretion in accordance with the risk of bleeding. The details of those strategies are presented in lines 98–103. The justifications for prescribing low dose rivaroxaban were as follows. First, the findings of J-ROCKET AF (15/10 mg/d) support the use of reduced-dose rivaroxaban as an alternative to standard-dose rivaroxaban or warfarin for stroke prevention in Asian patients. Second, in Asian countries, safety concerns based on the generally low body mass index of the population and their different susceptibility to bleeding have resulted in lower DOAC doses generally being preferred. Lastly, INR control can be difficult in Chinese patients. Patients undergoing LAAC were informed of the current indications and risks of low-dose rivaroxaban. Thus, low dose rivaroxaban was prescribed based on a patient’s financial status and willingness.

Point 3: Which device were sued over time and in both groups comparatively?

Response 3: Thank you for this question. We have added detailed data on the types of LAAC devices to Tables 1 and 2. There were no significant differences in type of LAAC device between the rivaroxaban and warfarin groups.

Point 4: Please provided not just TTR but INR range

Response 4: Thank you for your suggestion, in response to which we have revised the manuscript as follows. The sentence “Standard warfarin therapy was defined as taking warfarin for at least 45 days after LAAC with a target INR between 2 and 3 and an individual therapeutic range >70%” has been added on lines 116–118.

Point 5: CI are suggested. I am not so sure whether the conclusions would stand a larger trial. Therefore, more careful wording is suggested.

Response 5: Thank you for this suggestion. We have revised the manuscript accordingly, including adding CIs as necessary in the Results section (lines 158–190). Furthermore, we have also modified our wording in line with your comment. In particular, we have added the following. “…the limited number of cases and low incidences of DRT, ischemic stroke/transient ischemic attack/systemic embolism, and bleeding events after implantation may have prevented the differences between the groups from reaching statistical significance. Hence, larger prospectively designed studies are needed to provide high quality data on this topic” to the limitations section (lines 276–280).

Point 6: Table 3/4: how many patients were on OAC and APT? any information about OAC APT at time of event? Temporal distribution of events from time of device implantation might be helpful.

Response 6: Thank you for these questions. No patients had a combination of OAC and APT following LAAC. Following successful LAAC, OAC therapy (without APT) was usually continued for at least 45 days, and DAPT (aspirin 100 mg plus clopidogrel 75 mg) for 1.5–6 months. Lifelong SAPT was subsequently prescribed. All patients who had adverse events were on routine antithrombotic protocols. Tables 3 and 4 provide information about post-procedural antithrombotic strategies at the times of those events. For example, as shown in Table 3, four patients taking OACs had ischemic strokes 0–1.5 months after placing an LAAC.

Point 7: The paper would benefit from some rewording, in particular the part highlighted in yellow

Response 7: Thank you for this comment, in response to which we have revised the manuscript to the best of our ability. Moreover, Kelly Zammit, BVSc, and Dr Trish Reynolds, MBBS, FRACP from Liwen Bianji (Edanz; http://www.liwenbianji.cn/) have provided English language editing to minimize grammatical and other errors.

Reviewer 2 Report

The authors have investigated and important cardiovascular theme regarding anticoagulation in a specific subgroup of patients undergoing left atrial appendage closure. As of today guidelines in this specific contest are scarce and leave therapy evaluation completely to practitioner clinical experience. To this regard this study is of great interest to help future guideline orientation. 

Overall, the study is interesting, well thought of and well written.
Introduction is sufficient. The present table and the figure are of universal interpretation. 

My revision consists in the need in the introduction for a figure with the procedure described and in the conclusion there is a need for future prospective. This review should recommend a possible change for guidelines in this field of application. 

Author Response

Point 1: The authors have investigated and important cardiovascular theme regarding anticoagulation in a specific subgroup of patients undergoing left atrial appendage closure. As of today guidelines in this specific contest are scarce and leave therapy evaluation completely to practitioner clinical experience. To this regard this study is of great interest to help future guideline orientation. 

Overall, the study is interesting, well thought of and well written.

Introduction is sufficient. The present table and the figure are of universal interpretation. 

Response 1: Thank you for your supportive comments. We have revised the manuscript in keeping with your advice.

Point 2: My revision consists in the need in the introduction for a figure with the procedure described and in the conclusion there is a need for future prospective. This review should recommend a possible change for guidelines in this field of application. 

Response 2: Thank you for these suggestions, in response to which we have provided Figure 1, which is a study flow diagram. General and detailed descriptions of our procedures are presented in lines 77–82 and 133–143. Furthermore, we have modified the conclusions by adding future perspectives as requested. “The present findings are important for clinical practice because DOAC are being increasingly prescribed as a result of their favorable safety and efficacy profiles. Either reduced- or half-dose rivaroxaban may constitute an effective and safe alternative to warfarin in patients with NVAF who are at high risk of bleeding, providing a good balance between the incidences of thromboembolism and major bleeding events during endothelialization after LAAC. Future studies should specifically evaluate the optimal dose of rivaroxaban for balancing effectiveness and safety in Asian patients who have undergone LAAC.” (lines 287–294)

Round 2

Reviewer 1 Report

good to go, thank you for your revisions

Reviewer 2 Report

The authors have provided the changes requested. The article is now fit for publication.